

# Auroral breakup detection in all-sky images by unsupervised learning

Noora Partamies[1], Bas Dol[1,2], Vincent Teissier[3], Liisa Juusola[4], Mikko Syrjäsuo[1], and Hjalmar Mulders[2]

[1]The University Centre in Svalbard, Longyearbyen, Norway
[2]Eindhoven University of Technology, The Netherlands
[3]ENSTA Bretagne, Brest, France
[4]Finnish Meteorological Institute, Helsinki, Finland

**Correspondence:** Noora Partamies (noora.partamies@unis.no)

**Abstract.** Due to a large number of automatic auroral camera systems on the ground, the image data analysis requires more efficiency than what a human expert visual inspection can provide. Furthermore, there is no solid consensus on how many different types or shapes exist in auroral displays. We report the first attempt to classify auroral morphological forms by unsupervised learning method on an image set that contains both nightside and dayside aurora. We used six months of full-colour auroral all-sky images captured at a high-arctic observatory on Svalbard, Norway, in 2019–2020. The selection of images containing aurora was performed manually. These images were then input to a convolutional neural network called SimCLR for feature extraction. The clustered and fused features resulted in 37 auroral morphological classes. In the classification of auroral image data with two different time resolutions we found that the occurrence of eight morphological classes strongly increased when the image cadence was high (24 seconds), while the occurrence of 13 morphological classes experienced little or no change with changes in input image cadence. We therefore investigated the temporal evolution of the group of eight "active auroral classes". Time periods for which "active auroral classes" persisted for longer than two consecutive images with maximum cadence of six minutes coincided with ground-magnetic deflections and their occurrence was found clustered around the magnetic midnight. The active auroral onsets typically included vortical auroral structures and equivalent current patterns typical for substorms. Our findings therefore suggest that our unsupervised image classification method can be used to detect auroral breakups in ground-based image datasets with a temporal accuracy determined by the image cadence.

## 1 Introduction

Auroral displays exhibit a vast variety of different morphology and dynamics. Despite decades of research and an early morphological "template" for auroral evolution during substorms (Akasofu, 1964), there are still many unexplained structures and periods of evolution, as pointed out by Knudsen et al. (2021) in a recent review. With the fast development of observational capabilities, recent observations have also revealed new auroral (or their like) forms (e.g. Lumikot – McKay et al. (2019), Fragments – Dreyer et al. (2021), STEVE – MacDonald et al. (2018)), which emphasises how we still lack knowledge and understanding of auroral structures and, in particular, their relation to each other.





One of the first automatic image classification attempts by Syrjäsuo and Donovan (2004) detected arcs, patches and omega bands. They stated that for less than 10% of all auroral structures the form is "known" and can be named. This obviously hampers our skills to perform morphological classification for aurora in statistical sense and by means of supervised learning. The (known) structures typically included in supervised learning as the ground truth are arcs, patchy/diffuse, discrete, moon and clouds (e.g. Clausen et al., 2018; Kvammen et al., 2020; Sado et al., 2022). The automatic classification results are good with about 90% success rate, but some of these commonly used auroral classes are very broad, with "Arcs" being the only specific shape and the rest of the auroral morphology being grouped into "Discrete" and "Diffuse" aurora. Furthermore, as about 50% of the image data do not contain aurora (but rather clear skies, clouds, or Moon), classifying a combination of aurora and non-aurora classes makes the morphological part of the classification inefficient. To overcome the problem of classifying largely unknown auroral features an auroral arciness index was introduced to include all images containing aurora (Partamies et al., 2014). This method recognises all auroral structures, and gives them an index-like number between 0.4 and 1 based on the distribution and clustering of the brightest pixels in the images. This method has been used to characterise temporal evolution of selected known structures (e.g. poleward moving auroral forms of the dayside aurora by Goertz et al., 2022).

An unsupervised learning attempt of the dayside auroral structures by Yang et al. (2021) clustered auroral forms into two classes based on 4000 randomly selected images over five auroral (northern winter) seasons. The number of clusters chosen by the authors was based on visualisation of the feature vectors. As daytime aurora has previously been divided into four categories by human experts (e.g. Hu et al., 2009), the new unsupervised clustering results reopens the discussion on the unknown number of true clusters. In their study, the first cluster contained variable morphological features, such as arcs, patches and spots with high brightness and primarily afternoon occurrence, while the second cluster consisted of corona-type aurora of lower brightness and high occurrence rate around magnetic noon.

To the best of our knowledge, we report the first attempt on unsupervised learning on auroral image data that includes both daytime and nighttime aurora. Our practical application is to automatically identify auroral breakups in the image data. This is particularly important for locations like Svalbard archipelago. Surrounded by a highly conductive ocean, the ground magnetic measurements are typically contaminated by the magnetic contribution of ground-induced currents by about 50–70% (Juusola et al., 2020), making the traditional substorm onset detection methods on magnetic data less reliable.

## 2   Auroral image data

We use full-colour all-sky camera (ASC) data from Kjell Henriksen Observatory (KHO, 78.25°N, 16.04°E) on Svalbard in arctic Norway. Our ASC is a Sony $\alpha$7s mirrorless DSLR, which has been in operation since late 2015, but has also had comparable predecessors since 2008. The ASC raw data has a high pixel resolution (2832×2832). However, our analysis uses quicklook data with reduced pixel and time resolution for faster processing. Nighttime images with 4-second exposure



time have been taken at a cadence of 12 seconds throughout winter seasons, which on Svalbard extend from the beginning of November until the end of February. Images are captured when the Sun is below the horizon.

Daily summary plots of image data are automatically published in the form of keograms[1]. These are north-south slices of individual images stacked together into a daily time vs. latitude plots. Keograms have been used later on in the study to validate the results of our automatic classification.

In addition to the auroral image data, we use geomagnetic index data and solar wind data from OMNIWeb[2] to provide an overview of the magnetic activity level, as well as a reference for comparisons between what we call "active" and "quiet" auroral displays (for more detailed explanations, see section 4.1). Dst-index data from the World Data Center in Kyoto[3] is used as a storm indicator. Hp30-index[4] (Yamazaki et al., 2022) gives a 30-minute resolution version of planetary Kp-index, and the SuperMAG electrojet index (SML[5], Gjerloev (2012)) describes the auroral electrojet variability with a global coverage of magnetometer stations at 1-minute resolution.

## 3    Classification method

### 3.1    Pre-processing of the auroral images

Manual labelling was performed for all quicklook images from the winter season 2019–2020 as well as for January and February 2019. Quicklook images have a reduced resolution of 480×480 pixels and about 6-minute cadence. The manual labelling aimed at providing a ground truth for developing an automatic classification routine to detect images which contain auroral emission. This dataset contained approximately 37000 labelled images. From this dataset, only those images that contained auroral emission and were not dominated by clouds were further used in unsupervised learning. That became approximately 12000 images.

To prepare the images for unsupervised learning, each image was first processed to remove features that could lead to classification biases that are unrelated to the auroral morphology, such as a caption indicating the camera type, observatory location, date and time. After removing the caption, images with very faint or barely detectable aurora were removed. In the quicklook images, the colour of each pixel is represented by individual intensities in red, green and blue colours. This is commonly known as the RGB colour space. However, for processing based on brightness, we transform the RGB colours into HSV and L*a*b* colour spaces, where the brightness and actual colour content are more clearly separated.

In HSV-colour space, the colours are expressed by Hue (H) and Saturation (S) while the brightness is contained in Value (V) (Malacara, 2002). If V was less than 50 out of 255 in more than 90% of pixels in the image, the image was discarded as too faint. This left 10300 images containing sufficiently bright aurora.

---

[1]http://kho.unis.no/Keograms/keograms.php

[2]https://omniweb.gsfc.nasa.gov

[3]https://wdc.kugi.kyoto-u.ac.jp/dstdir/index.html

[4]https://kp.gfz-potsdam.de/en/hp30-hp60

[5]https://supermag.jhuapl.edu/indices/



The L*a*b*, or CIELAB colour space, aims to represent the colours as a human observer would perceive them (Malacara, 2002). Here, L* denotes the lightness of the colour, while a* and b* provide the colour information. Unlike RGB and HSV, L*a*b* is a device-independent colour space based on a standard human observer and a standard outdoor illumination. Similarly to earlier work (e.g. Sado et al., 2022; Clausen et al., 2018; Johnson et al., 2021), we clipped the L* values to the range from 0.5 to 99.5 percentiles. To reduce the influence of the background sky conditions on the aurora classes, the median values of the a* and b* colour channels were used to provide a neutral white balance for all images. Next, the images were cropped to 400x400 pixels around the centre. This removes the biases due to dark corners outside the circular field-of-view of all-sky images. This step also removes most of the auroral emission at the lowest elevation angles, which is beneficial as the morphology of the auroral forms near the horizon is heavily distorted by the fish-eye optics, and therefore difficult to examine.

To better classify faint aurora a contrast enhancement was performed on the images by equalizing the histograms of the L* channel for each image. Furthermore, a 5×5 median filter was used to remove single bright pixels due to stars, similarly to the 3×3 filter used by Kvammen et al. (2020). Finally, the images were resized to 224×224 pixels and converted back to RGB, which is the expected format for most feature extractors.

The cadence of 6 minutes of the quicklook data favours quiet aurora over active aurora. This happens because active aurora occurs in short highly dynamic bursts, which are poorly sampled by a cadence of 6 minutes, while quiet aurora experiences longer lifetimes. To study the influence of this bias and to better represent the active auroral forms, a second dataset was made. In this dataset, the cadence was reduced to 24 seconds during selected periods of active aurora. This additional dataset is also referred to as high resolution dataset in this work. These selected periods were visually identified as the 19 brightest auroral displays with large north-south extent as seen in the keograms (sample keogram in the top panel of Figure 2).

## 3.2  Classification of images

In contrast to *supervised learning*, where we know the correct answer to a classification question and can train a classifier accordingly, in *unsupervised learning* we do not know the answer. The manual labelling of auroral images provided us with a dataset of unknown auroral forms. We try to learn the types of the aurora from the data by using clustering algorithms. Intuitively, similar auroral shapes in the images should belong to the same cluster.

The first step is to define a numeric measure of the image content in form of a *feature vector*. In the second step, the images are clustered based on their feature vectors: here, the true number of clusters is, of course, unverifiable. As clustering is often based on similarity metrics, the results also depend on how well the feature vectors represent the image content. In his Master's thesis, Teissier (2022) evaluated a number of different feature vector extractors as well as clustering approaches. The following provides a brief description of the central concepts and the choices of methodology.

Classifiers based on Convolutional Neural Networks (CNN) (LeCun et al., 2015) are becoming increasingly popular in image analysis. *Training* an artificial neural network refers to the process of optimising all parameters within the network to minimise the classification error.

The analysis of auroral images in our dataset uses a processing chain which includes a CNN architecture followed by a smaller neural network. We start with the inclusion of high-time resolution data as it provides a better balance between images



of auroral display during active and quiet times. We use a variation of the SimCLR feature extractor used by Johnson et al. (2021). This feature extractor builds on the Resnet-50 neural network (He, et al., 2016) architecture. Random rotations and other image transformations are often used in training image classifiers to both artificially increase the number of "different" sample images and to obtain invariance in orientation. In this study, we limit ourselves to horizontal flipping (east-west in auroral images) and random cropping (random parts of the sky with aurora) in the training phase (Teissier, 2022).

We used the UMAP (Uniform Manifold Approximation and Projection for Dimension Reduction) method to reduce the dimensionality of the feature vectors (McInnes et al., 2020). This step was carried out to improve clustering (Steinbach et al., 2004). In the UMAP reducer, there are two critical parameters: one that impacts on how much the algorithm focuses on local versus global structures (N_neighbours), and another determining the minimum allowed distance between features in the dimensionally reduced space (min_dist). Based on the experiments by Teissier (2022), we used the values N_neighbours = 20 and min_dist = 0.

Finally, a clustering algorithm was used to determine clusters in the dimensionally reduced feature vectors. Two different clustering algorithms were used: K-means (MacQueen, 1967) and hierarchical clustering (Nielsen, 2016). These algorithms were used to divide images into 30 or 100 clusters. Consequently, the four clustering results (two clustering algorithms and two numbers of clusters) were fused together to balance the weaknesses and biases in the individual approaches. This also helps to identify outlier images that are connected to different auroral features depending on the clustering algorithm that is used. The fusion was done by using a modified majority voting with co-association matrix algorithm (Fred, 2001). The original algorithm was modified to have two tuneable parameters. The first parameter is the majority threshold of the co-association matrix, above which two samples are merged in the same cluster. The second parameter is the minimal cluster size, under which a cluster is merged into the cluster of 'outliers', to indicate hard to classify images. The values of these parameters were determined by changing them to minimize the loss-function

$$L = \frac{1}{n_c} \sum_{k=0}^{N} |n_k - n_0^k| + \frac{n_e}{n_c} \quad \text{with } n_0^k = \frac{N}{n_c} \text{ if } k < n_c \text{, otherwise } n_0^k = 0 \tag{1}$$

where $N$ is the total number of samples, $n_k$ the number of samples in the cluster $k$, and $n_e$ the number of outlier samples. The parameter $n_c$ is used to influence the number of clusters resulting from the fusion, and was set to 25. Image classes were determined by the cluster to which an image feature vector belonged. The fusion resulted in 37 classes with class numbers ranging from -1 to 35, where the -1 class is for the outlier images not belonging to any of the other classes.

Using a basic desktop computer (in 2022), the pre-processing, training and clustering required roughly one day of computer time. The classification based on determined clusters took approximately 45 minutes for 9200 auroral images, or less than one second per image. Table 1 summarises the numbers of images at each stage of the data processing.



**Table 1.** Summary of numbers of images in our datasets

| Dataset | Number of images | Notes |
|---|---|---|
| Raw data | 37000 | All labelled images, 6-min cadence |
| Aurora only | 12000 | Images with no aurora removed |
| High-time resolution events | 5500 | Additional bright aurora events with 24-s cadence |
| Clustered aurora | 9200 | Auroral images grouped by their morphology |

## 4 Results

### 4.1 Differences between active and quiet type aurora

The CNN analysis was run on the quicklook images with 6-minute cadence alone as well as on the full dataset with the inclusion of higher resolution images with 24-second cadence. As a result we found that the occurrence of some morphological classes increased much with the increasing temporal resolution, while the occurrence rate of other classes was not affected. The high sensitivity to the increased temporal resolution of the image data (at least 80% change, red line in Figure 1) therefore suggests that these classes primarily describe morphology of active auroral displays. These are classes number 0, 13, 26, 8, 21, 7, 14

and 35. The class occurrence changes are illustrated by Figure 1, which shows the occurrence rate difference of all 37 classes between quicklook and high-resolution data in percentage with respect to the total occurrence rate of each class. The classes, for which the occurrence showed little or no change (maximum 20% change, blue line in Figure 1) with the increasing temporal resolution (classes number 20, 4, 9, 17, 34, 22, 32, 10, 3, 18, 25, 1 and 29), are likely to contain auroral structures mainly related to quiet auroral displays. By quiet auroral displays we refer to images of structures, which are relatively simple and

evolve slowly in time. A typical example of this is an auroral arc, or a multiple arc, which changes little between consecutive images with higher temporal resolution of 24 seconds. Example images of slowly evolving features can be seen in the bottom row in Figure 3. Active auroral displays, in turn, refer to images with one or multiple more complex auroral structures, which evolve significantly between consecutive images in our higher temporal resolution dataset, and have little resemblance between consecutive images in quicklook time resolution (6 minutes) as seen in the top row of Figure 3. These active auroral displays

typically include bright emission in vortical and small-scale structures. The threshold values for the occurrence rate changes for quiet and active aurora (20% & 80% respectively) are based on visual inspection of the class occurrence rate histogram in Figure 1, and are therefore somewhat arbitrary. The choices were made with the idea that large percentage difference is a conservative choice as long as it results in several classes in each extreme.

An example of time evolution of auroral all-sky image data with the detected active and quiet auroral displays is provided by

Figure 2, where the upper panel is a full keogram of 25 January 2019 and the lower panel shows a time evolution of individual image labels of active (red) and quiet (blue) auroral displays. Each individual label is marked as a dot. The labelled images are clustered at the time periods of clear skies and aurora, because our pre-processing excludes clear skies with no aurora and images with clouds. The active aurora clusters coincide with the brightest aurora in the keograms, as expected. The bottom

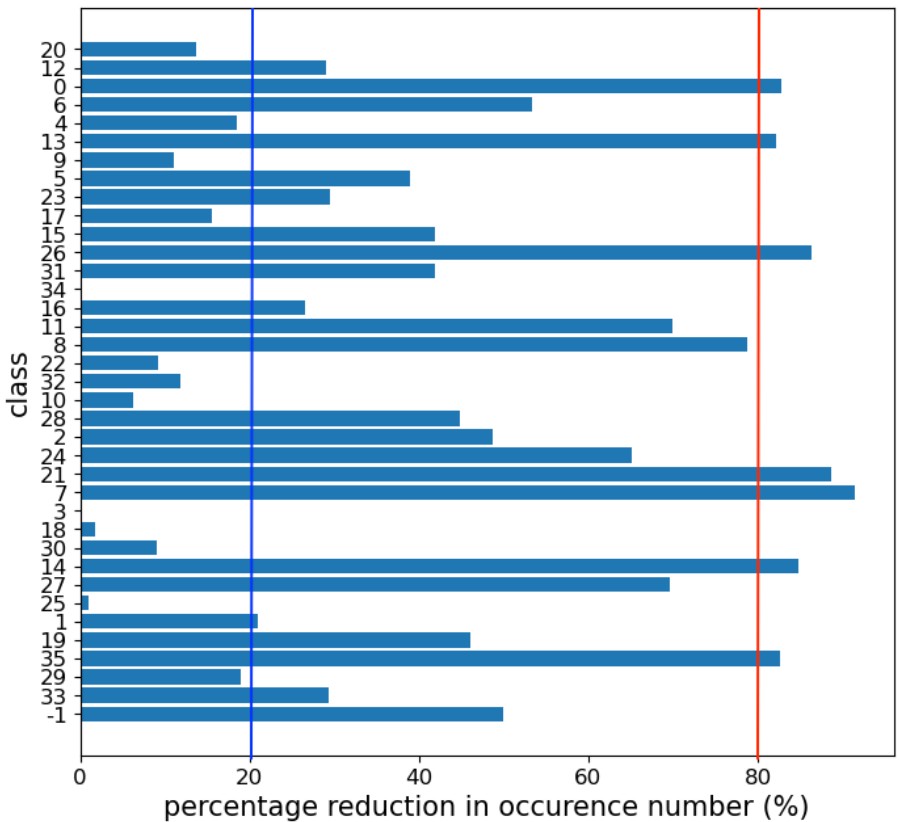

**Figure 1.** Occurrence rate difference of all 37 classes between quicklook and high-resolution data in percentage with respect to the total occurrence rate of each class. Classes with maximum 20% change are called quiet aurora, and classes with at least 80% change are taken as active aurora. Threshold occurrence changes for quiet and active aurora are marked by blue and red vertical lines respectively. Class number -1 is an outlier class.

panel in Figure 2 shows ground magnetic data (X component) from Svalbard magnetometer stations in Hornsund (HOR, 77.00°Glat), Longyearbyen (LYR, 78.20°Glat) and Ny-Ålesund (NAL, 78.92°Glat). These stations are part of the IMAGE magnetometer network[6] (Tanskanen, 2009). The magnetic variations are measured and plotted in 10-second time resolution. The three periods of active aurora correspond to small (∼100 nT) to moderate (∼400 nT) magnetic deflections in the north–south component. They could all be interpreted as substorms, the first of which expands towards Svalbard from further south, while the two latter events occur more directly overhead. The visual correspondence between the periods of active aurora and the magnetic deflections during the two latter substorms appears particularly good.

To further investigate these difference morphological class groups of "active" and "quiet" aurora, we selected time periods of active or quiet classes that included three or more consecutive images with a maximum cadence of 6 minutes. For a sequence longer than 3 consecutive images, one single quiet class image would be allowed within an active auroral period, or the other

---

[6]https://space.fmi.fi/image/www/index.php



**Figure 2.** Top: Sample keogram of colour ASC data on 25 January 2019. At 00-01 UT on the southern sky (at scan angles 160–180) the moon is visible and illuminates the clouds in the field-of-view. At 03–08 UT faint morning sector aurora is seen, first as green and later also as red. At 08–15 UT the daylight increases the illumination of the sky although the sun is below the horizon. Due to the increased light level the exposure time changes, and that is seen as vertical colour changes at about 09 and 13:30 UT. From 15 UT onwards the sky is dark and green aurora appears in three northward expansions. Middle: Time series of active (red) and quiet (blue) class labels for the sample day. Bottom: Evolution of ground magnetic X-component deflections at the stations of NAL (yellow), LYR (red), and HOR (blue). LYR data are missing until 14 UT, but exists during the active aurora displays.





**Figure 3.** Top: Examples of active auroral displays from the auroral activation at about 19 UT on 25 January 2019. Images are the first (18:38:45 UT, class 0), middle (18:59:21 UT, class 7), and last one (19:19:27 UT, class 7) labelled as active aurora. Bottom: Examples of quiet auroral displays at in between the two largest auroral activations on 25 January 2019. Images are the first (19:21:25 UT, class 9), middle (19:31:05 UT, class 32) and last one (20:18:02 UT, class 32) labelled as quiet aurora. In the images the magnetic north is to the top and east to the left.

way round. This procedure resulted in 28 active auroral events and 222 quiet auroral events in the time frame of all our labelled
data in 2019–2020.

The table 2 gives an overview of the typical parameters for periods of active and quiet aurora. The duration of the active aurora events varied from 60 seconds up to 5 hours, while that of the quiet events ranged from 2 minutes up to 25 hours. The median values (row 2 in the Table), however, do not differ much. The active aurora events typically started at magnetic midnight (row 3), while the quiet aurora events were spread more evenly over the Magnetic Local Time (MLT) hours, which
is illustrated by Figure 4. The median start time is therefore not a good indicator for quiet aurora, which experiences three occurrence maxima in 24 hours: right after midnight, at about 6–9, and at about 15–19 MLT. The geomagnetic indices with



**Table 2.** Median values of event durations, geomagnetic indices and solar wind speed for periods of active and quiet aurora, respectively.

| Parameter | Active aurora | Quiet aurora |
|---|---|---|
| 1. Number of events | 28 | 222 |
| 2. Duration (min) | 37 | 42 |
| 3. Start time in MLT | 0 | 10 |
| 4. Dst index (nT) | -7.0 | -4.0 |
| 5. SML index (nT) | -180 | -50 |
| 6. Hp30 index | 2.0 | 1.33 |
| 7. Solar wind speed (km/s) | 416 | 367 |

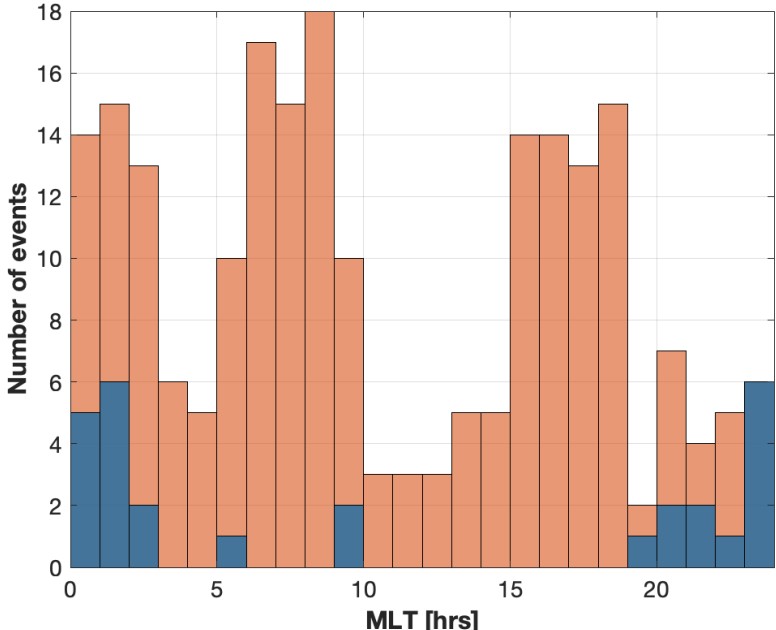

**Figure 4.** Distribution of start times of active (blue) and quiet auroral displays (orange) in MLT. Midnight in MLT is at about 21 UT on Svalbard.

coarser spatial resolution (a low number of stations, which provide the data) are not dramatically different between the periods of active and quiet aurora. These are the Dst index (row 4) and the Hp30 index (row 6). Neither of these indices with low spatial resolution includes high-latitude magnetic field data, and therefore they do not reflect the high-latitude magnetic deflections, which may have different timing compared to the corresponding deflections at lower latitudes. However, the SML electrojet index (row 5), which has a good spatial coverage of magnetic data from a range of different latitudes, shows a difference of





more than 100 nT between active and quiet aurora. Similarly to the low resolution magnetic indices, the median solar wind speed (row 7) is more enhanced during active than quiet aurora, but only some tens of km/s, which is not significant.

## 4.2 Are the active aurora periods auroral breakups?

We have performed a visual inspection on what the active aurora periods correspond to by using keograms (as in top panel in Figure 2). The main mission of the validation procedure was to find out what the detected events looked like, if they corresponded to magnetic substorm activity, could keogram inspection reveal missed auroral activations, and if yes, what kind of events they would be. Keograms were therefore viewed for each day from January 2019 until March 2020. We inspected all detected active aurora events, as well as each keogram for any additional auroral brightening, which was not detected.

Most active aurora periods were auroral brightenings for which the bright emission extended over the full (or nearly full) field-of-view of the all-sky image as illustrated by the keogram in the top panel of Figure 2. Most of the active aurora periods also coincided with a small to moderate deflection of the local magnetic north–south (X) component, which indicates substorm activity. The maximum magnetic deflections for the active aurora events ranged from about 50 nT up to about 400 nT, with a median value of 100–200 nT. Individual image labels were also investigated during the active aurora periods, but no consistent

time evolution or preferential order was found among the individual active aurora classes.

Taking the auroral breakup overhead as a benchmark, five false positive events were identified in the visual inspection. Three of our false positive cases were morning sector auroral events (see the lonely blue bars in Figure 4). These consisted of transient brightenings in the morning sector diffuse and pulsating aurora. Two of these events are from the shortest end of the duration spectrum containing only two consecutive images. This type of events could be filtered out simply by increasing the minimum

duration of the active aurora events in order to produce a cleaner auroral breakup list. Two of the other events we call false positive cases were clearly substorm-like aurora, which extended about half way of the sky towards the local zenith, such as the first active aurora event in Figure 2. This type of events are not exactly wrongly detected but rather a milder category compared to the rest of the cases. They may include substorms, which are inherently not taking place at high latitudes but rather expand towards Svalbard from further south where the strongest magnetic disturbances take place.

As for false negatives, i.e. any additional auroral brightenings that the active aurora periods did not cover, no events were found. Based on keograms, we visually identified some auroral brightenings that extended nearly across the full sky but were not automatically labelled as active aurora. These included 1) events where the sky was fully or partly cloudy so that the auroral structures were obscured, 2) events where the image was contaminated by moonlight so that the auroral brightness (and therefore the structure) was not well visible, or 3) events, which were substorm-like auroral activations that did not even reach

half way toward the local zenith. Images in the two first categories would have been excluded from the data fed to the classifier due to the clouds or the moonlight. In the last category, the largest magnetic deflections occurred typically further south from Svalbard, which was confirmed by visual examination of the IMAGE magnetometer data.

Figure 5 shows snapshots of the ionospheric equivalent currents, derived from 10-second IMAGE magnetometer data (Van-

hamäki and Juusola, 2020), on 25 January 2019 around a period of active aurora in Figure 2. The plots show an equivalent





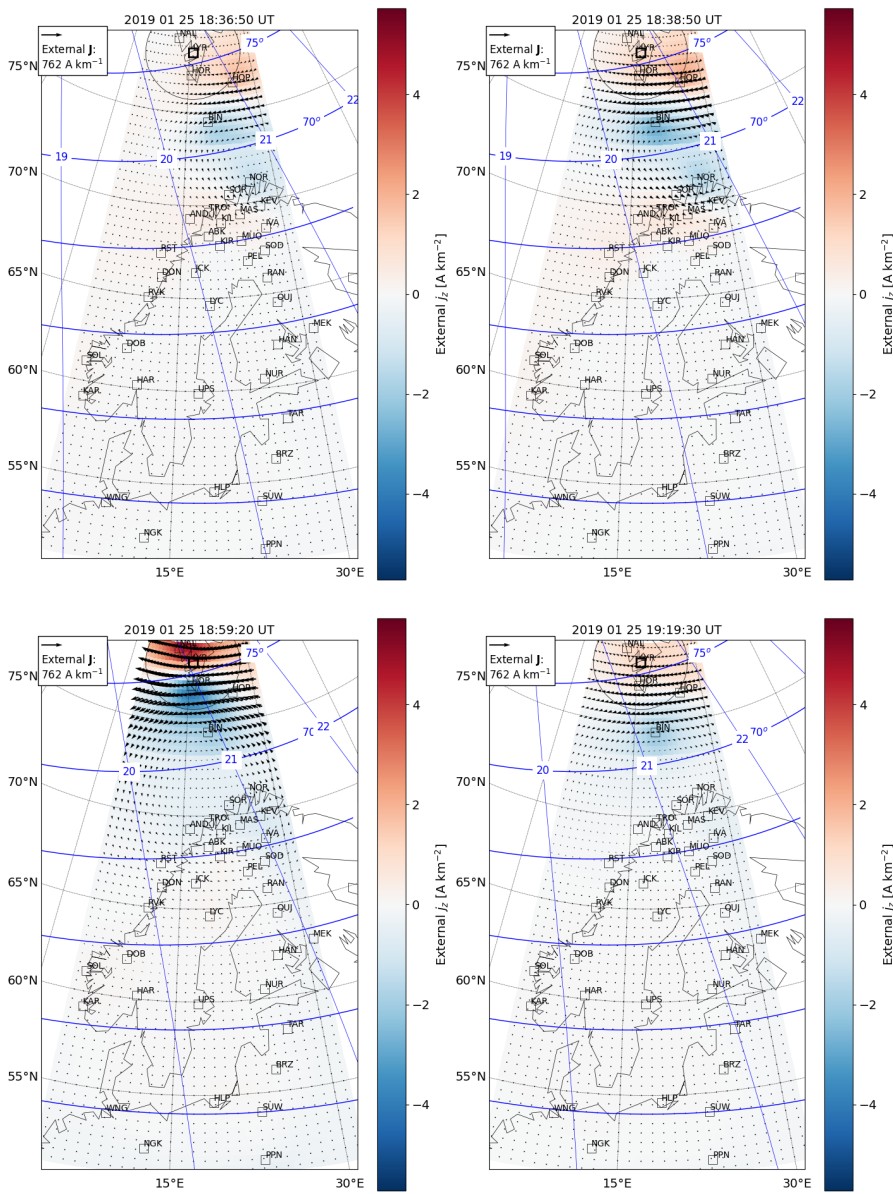

**Figure 5.** Ionospheric equivalent currents (Vanhamäki and Juusola, 2020) (arrows) and their curl (color) on 25 January 2019 two minutes before a period of active auroras started (18:36:50 UT), at the start of the period of active auroras (18:38:45 UT), during the period of active auroras (18:59:20 UT), and at the end of the period of active auroras (19:19:27 UT). The IMAGE magnetometer stations used to construct the equivalent currents are shown by black rectangles and LYR field-of-view by a circle. Quasi-dipole (QD) coordinates (Richmond, 1995; Emmert et al., 2010; Laundal et al., 2022) are indicated by the blue grid.




current pattern typical for the Westward Traveling Surge (WTS) (e.g., Vanhamäki et al., 2005) sweeping past LYR from south-east to northwest. Two minutes before the period of active aurora (18:36:45 UT), the U-shaped head of the surge pattern was still located south and east of the LYR field-of-view (indicated by the black circle). The surge head appears to reach LYR at the time the active aurora started (18:38:45 UT) and to disappear around the time the active aurora ended (19:19:27 UT). The

closest auroral images to the start and the end of the active aurora can be seen in the top row of Figure 3. A similar sequence of equivalent current dynamics (not shown) can be seen around the next interval of active aurora in Figure 2, between 20:18:14 and 20:41:04 UT. We have also examined several other active aurora events and found similar equivalent current development. This indicates that periods of active aurora may be associated with a WTS, which has originated earlier at a substorm onset location just east and south of LYR, and travels westward and northward past LYR.

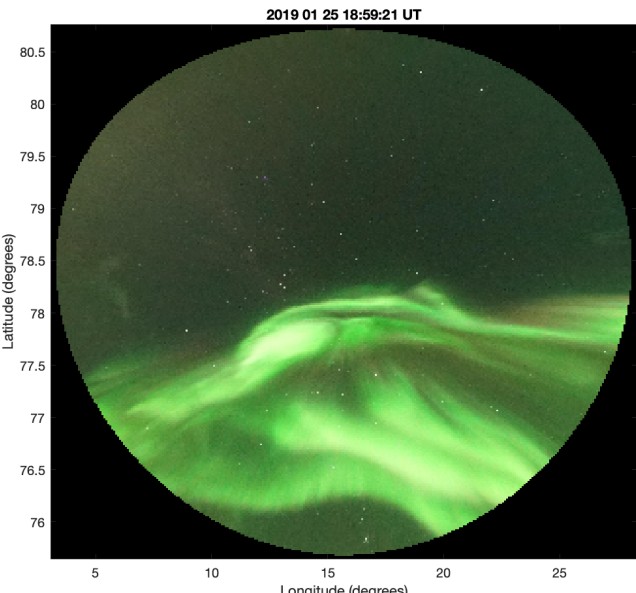

**Figure 6.** An image of the surge head taken at 18:59:21 UT (same as the middle panel in the top row of Figure 3) plotted into geographic coordinates for easier comparison with the corresponding distribution of equivalent currents in Figure 5. The assumed emission peak emission height for the mapping was 110 km.

Figure 6 shows the ASC image of the surge head aurora as it comes into the ASC field-of-view, which corresponds to the strongest ionospheric equivalent currents at 18:59:20 UT (Figure 5). This is the same image as the middle one in the top row of Figure 3, but here it is plotted onto geographic coordinates to make it more comparable to the coordinates of the equivalent current maps (grey grid). Characteristic to the WTS aurora is the intense auroral emission and vortical structures, located within the region of upward field-aligned current (blue colour, negative external $j_Z$). The auroral evolution leading to this included

poleward expansion from the southern edge of the field-of-view, as well as more localised brightenings propagating from east



to west along the poleward boundary of the surge aurora. This evolution is in agreement with that of the equivalent currents as described above.

## 5 Discussion

Results from our newly developed method for unsupervised classification of auroral images into morphological classes have
been used to detect auroral activations over Svalbard. While the classification method produced 37 individual classes, our further analysis suggests that the occurrence of eight of those classes are associated with active auroral displays. Similarly, 13 classes are prominently present during quiet auroral displays. Particularly interesting finding is that longer periods of active auroral displays practically include all full-sky auroral events in our all-sky field-of-view. These events coincide with ground-magnetic deflections and enhanced westward ionospheric electrojet current, which are signatures of magnetospheric substorms.
At high-latitudes and close to the polar cap boundary, such as Svalbard is (78° Glat), auroral breakups overhead do not happen often. It is therefore practical to have a way to automatically detect them in the image data, in such a way that no events are missed, even when this means a detection of some additional false positive cases (10–15%).

The MLT distribution of quiet aurora in Figure 4 has three maxima: one at around midnight, one in the afternoon at 15–17,
as well as one in the moring at 6–9 MLT. This is in a very good agreement with an MLT distribution of auroral structures for Svalbard during quiet geomagnetic conditions (Partamies et al., 2022). In their 13-year long data series the auroral morphology was investigated in terms of simple auroral structures called auroral arcs versus other more complicated structures. Both classes undergo similar MLT occurrence rate when geomagnetic conditions are quiet (auroral electrojet AL index is larger than -100 nT). These quiet conditions correspond to the magnetic disurbance level for our quiet aurora. Similarly, for their
active conditions (AL smaller than -300 nT), the MLT occurrance of both arcs and more complex auroral structures maximise between 21 and 02 MLT. This agrees with the occurrence of our active aurora, even though our active aurora events generally take place in less disturbed conditions (SML median of -180 nT).

An earlier study by Signh et al. (2012) analysed magnetic signatures of high-latitude substorms poleward of the central auroral oval. They concluded that the substorms occur primarily close to the MLT midnight, at 21–02 MLT during low or
moderate solar wind stream, which is in a very good agreement with the occurrence conditions of our events. In a more recent high-latitude substorm study Cresswell-Moorcock et al. (2013) identified 112 events of energetic electron precipitation (EEP) in 12 months of electron density profiles from EISCAT Svalbard radar (ESR) measurements. This is a much larger number than what we collected in our study, but since our optical approach requires both dark and clear skies (possible only for three months a year) the event numbers seem comparable. Our events were collected from a total of six months of data, and sta-
tistically about half of the auroral images are cloudy. Our events of active aurora show a very similar clustering around the magnetic midnight as do the EEP events detected in radar data (their Figure 3a). Our auroral breakups occurred during less active magnetic conditions as compared to the EEP events from the radar data (Kp/Hp index = 2 for ours and 3 for theirs), and during average solar wind speed, while the EEP events were reported to take place during fast solar wind. It is very likely that



our auroral breakups would therefore not fulfill the EEP criterion of strong D region ionisation.


An automated method, which would closest compare to our approach is a new study on pixel level classification that has empirically implemented thresholds for detecting auroral intensifications by Yamauchi et al. (2023). This method includes auroral breakups among other brightenings. What the authors call a local-arc brightening is not validated strictly as substorm activity, because brightenings take place at all local times. The pixel-level classification of a full-resolution ASC image requires

a lot more manual work than manual labelling of full images, but since the authors used data from the same camera model as ours, a cross-validation of our results can be performed in the future.

ASC image data have been used to detect the auroral breakups before. Murphy et al. (2014) developed a method which utilised the temporal evolution of the auroral image brightness as a proxy for auroral breakups. Three independently studied substorms were identified with good accuracy. Furthermore, 50% of independently listed 240 substorm onsets were detected

within the uncertainty of their method.

Currently, the only operational image classification method that runs on real-time image data is described by Nanjo et al. (2022). They use un-pruned auroral colour images from Tromsø and Skibotn in Northern Norway and Kiruna in Northern Sweden[7]. The closest to the auroral breakup detection of that approach is the beginning of an extended period of the class called discrete aurora. The class itself contains much more than just the auroral breakups, but with some further analysis those

results may also become a helpful cross-validation dataset for our method in the future.

While the results presented here are very promising, further research is needed to assess how efficiently this method works on unseen image data from different winter seasons. This essentially depends on how well the originally manually labelled image data from 2019–2020 represents all different sky conditions and the variety of auroral morphology at and around the polar cap boundary. Future studies will also include testing the method on other similar ASCs at different locations. This may require

re-evaluation of the active and quiet aurora classes, as individual instruments have slightly different colour balances, which may affect the classification results. Furthermore, aurora at lower latitudes consists of different structures to some extent, and will be limited to the nightside aurora with different emission balance and different background sky conditions as compared to the dayside aurora. All these factors may affect the occurrence of the morphological classes, and therefore must be investigated. The traditional way of detecting substorms is to use measurements of ground-magnetic field. Because the magnetic data is not

sensitive to weather or daylight conditions, the magnetometer station density is high, and the data are available in long-time series in high temporal resolution, the magnetic approach has been widely used for statistical studies (e.g. Forsyth et al., 2015; Partamies et al., 2013; Juusola et al., 2011). However, the recently documented ground-induction contribution in the magnetic measurements suggests that the magnetic substorm detection may be biased (Juusola et al., 2020). Furthermore, for some studies it is essential to include information on the auroral morphology, which makes the availability of the image data a key

element.

---

[7]https://tromsoe-ai.cei.uec.ac.jp/

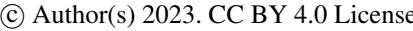


## 6 Conclusions

This study has explored a newly developed prototype method for automatic classification of auroral all-sky images in an unsupervised way. We used manually labelled data that were known to contain aurora, which means that in order for this approach to work, another classification of raw images into classes of Aurora and No Aurora is needed. The classification produced 37 classes. Our results showed that an occurrence of a subset of classes was strongly increased with an increasing temporal resolution of input images taken during bright auroral displays. This indicates that a time sequence of these morphological classes mainly describes auroral activations. This property was further used to detect the start times of periods with continous auroral activity in our labelled data. Examination of the detected auroral activations showed that these are indeed local auroral breakups that carry substorm-like features to the extent high-latitude substorm events are expected to. An essential property of our method is that no major events were missed by the detection. These are promising results that may help in identifying optical auroral breakups in the future, after being tested on unseen data and cross-validated with other methods.

*Code availability.* Unsupervised image classification method by Vincent Teissier: https://github.com/Tadlai/auroral-classification ; Master's thesis on unsupervised image classification by Vincent Teissier: http://kho.unis.no/doc/Vincent_master_thesis-final.pdf . The software to calculate the equivalent currents is available as supplementary material of Vanhamäki and Juusola (2020).

*Data availability.* Labelled images and their morphological classes are available at http://doi.org/10.11582/2023.00132 . IMAGE magnetometer data are available at
https://space.fmi.fi/image/www/index.php, Sony keograms are available at http://kho.unis.no/Keograms/keograms.php

*Author contributions.* Noora Partamies has supervised the projects by Vincent Teissier and Bas Dol. She has further analysed the classification results and taken the main responsibility for writing the manuscript. Vincent Teissier has developed the unsupervised learning method for his M.Sc project, helped Bas Dol to reproduce the results and further develop the method, as well as helped in describing the method in this manuscript. Bas Dol has produced the preliminary results on the unsupervised auroral classification, which were the starting point of this manuscript. Mikko Syrjäsuo has co-supervised the method development by Vincent Teissier and participated in data processing, visualisation and analysis of the classification results. Hjalmar Mulders has supervised the project by Bas Dol and participated in the brainstorming of the analysis. Liisa Juusola has analysed the equivalent currents for the periods of active aurora, plotted and interpreted the data. All authors have helped in editing the manuscript.

*Competing interests.* The authors declare no competing interests.



*Acknowledgements.* We thank the institutes who maintain the IMAGE Magnetometer Array: Tromsø Geophysical Observatory of UiT the Arctic University of Norway (Norway), Finnish Meteorological Institute (Finland), Institute of Geophysics Polish Academy of Sciences (Poland), GFZ German Research Centre for Geosciences (Germany), Geological Survey of Sweden (Sweden), Swedish Institute of Space Physics (Sweden), Sodankylä Geophysical Observatory of the University of Oulu (Finland), DTU Technical University of Denmark (Denmark), and Science Institute of the University of Iceland (Iceland). The provisioning of data from AAL, GOT, HAS, NRA, VXJ, FKP, ROE, BFE, BOR, HOV, SCO, KUL, and NAQ is supported by the ESA contracts number 4000128139/19/D/CT as well as 4000138064/22/D/KS.



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
