# Peer review of "Auroral breakup detection in all-sky images by unsupervised learning"

_EGUsphere, 2023_

## Author Response (AR1)

We thank both referees for careful reading of the manuscript and valuable comments. We have done our best to take into account and implement all suggestions, and we feel that they greatly improve the manuscript. Below the comments are copied in bold and our answers in regular font.

Comments from Referee #1:

**The classes found are not described in detail. Although this would take up space (likely quite a bit of it), if there is value in the classification it should be better described. The lack of description does not allow comparative study by others, nor comparison with previously established classes which are mentioned.**

The individual clusters are not described in detail because they result from an unsupervised clustering based on automatically determined image features. They do not form any obvious groups of auroral structures with respect to what we know from before, but based on visual inspection properties like contrast, brightness, colour, alignment and the location of the aurora in the images may play a role. It is therefore not straightforward to compare the content of these individual clusters to earlier human classified or supervised learning results. To make the distinction clearer between a pre-determined class (ground truth) and a numerically calculated cluster we now call the unsupervised method clustering and the results 37 individual clusters.
The results of this study are not dependent on understanding of the individual clusters. We merely investigate changes in the occurrence of cluster groups as a function of the temporal resolution of the input image data.

Cluster groups called active and quiet aurora are determined by their occurrence rate change as a function of input data cadence. A better description of the category of active (8 clusters) and quiet aurora (14 clusters) will be included in the revised version of the manuscript to say that the active aurora mainly includes large vortex structures, very bright or rayed arc-like structures. The quiet aurora primarily includes arc-like and multiple arc-like structures, faint diffuse aurora, and daytime and afternoon overhead aurora (corona) with a notable red emission component. Images with moonlight are grouped into the same clusters in

each category, although the auroral structures in those images are similar to those in other clusters within the same category. Instead of the old Figure 4 with a few examples of active and quiet aurora, we include figures with 10 random images of each individual cluster in the category of quiet and active aurora to support the description above.

*Technical corrections:*

**The article is well written but use of the word "the" should be reviewed, especially in the abstract. There is repetition of "https://doi.org" in some of the references. Arrowheads in Fig. 5 are difficult to see, and the caption refers to curl (which is likely correct as plotted) whereas the color bar label refers to Jz which is basically an assumption of the inversion method.**

The article has been reviewed and some words "the" removed. The reference list has been reworked into a more coherent list with no repeating doi.org's. Figure 5 has been replotted with cropped area for the maps, which makes the arrowheads better visible. The new colour bar label is Z component of the curl of the equivalent current. We use the same terminology in the caption, but also mention that this vertical component of the curl of the equivalent current can be used as an estimate of the field-aligned current, which is what is done in the text later on.

————

Specific comments from Referee #2:

**Line 93: It is possible that normalizing the brightness of each image could be introducing biases.  For example, faint arcs could then get lumped together with much brighter ones, which may not be a good thing for categorizing the aurora.  Perhaps this could be mentioned in the discussion.**

Sounds reasonable, the new version will mention in the discussion that it

should be investigated further if the brightness normalisation leads to unnecessary high weight on faint auroral structures as compared to bright auroral structures.

**Lines 75 to 95: It might be useful to add an example image, starting with the original All sky image, then how it looks at the different stages of this processing.**

Good point. Rather than showing each stage of the pre-processing of the images, some of which are visually minor, we include a figure with a couple of examples of the "raw" quicklook images prior to the pre-processing and the corresponding images after the pre-processing in the new version of the manuscript. They help illustrating the effect of cropping, colour enhancement and brightness normalisation.